- 1 Technical note: Water vapor sampling for the analysis of water stable isotopes
- 2 in trees and soils optimizing sampling protocols

8

- Alberto Iraheta<sup>1\*</sup>, Elise Malsch-Fröhlich<sup>1</sup>, Malkin Gerchow<sup>1</sup> and Matthias Beyer<sup>1</sup>
- \*Correspondence to: Alberto Iraheta (a.iraheta@tu-braunschweig.de)
- <sup>1</sup>Institute of Geoecology, Technical University of Braunschweig, Braunschweig,
- Germany

#### Abstract

Water isotopes are fundamental to ecohydrology, tracing water fluxes across the 10 continuum from soil to plants and the atmosphere. Most methods rely on destructive 11 sampling of water in soil and plant material. In recent years, techniques for collecting 12 equilibrated water vapor have been developed as a semi-in situ alternative. Here, we 13 present a systematic evaluation of water vapor sampling methods to identify how 14 storage time, flow rate, container type, and temperature influence the isotopic stability 15 of  $\delta^{18}$ O and  $\delta^{2}$ H. Controlled laboratory experiments tested storage times of 6, 24, 72, and 168 hours; temperatures of 4 °C, 20-25 °C, and 40 °C; flow rates of 35, 75, 100, 16 17 and 125 ml/min; and three container types: 250 ml infusion glass bottles (ND32), 1 L 18 FlexFoil® sample bags, and 500 ml Aluminum-Zip bags. Glass bottles showed the 19 highest isotopic stability, with deviations within  $\pm 0.5\%$  for  $\delta^{18}O$  and  $\pm 1\%$  for  $\delta^{2}H$  during 20 the first 24 hours. In contrast, FlexFoil® bags exhibited variability of  $\pm 1\%$  for  $\delta^{18}O$  and 21 -3 to +2% for  $\delta^2$ H, while aluminum-zip bags showed the largest offsets (-2.5 to -3% 22 for  $\delta^{18}$ O and -12 to -25% for  $\delta^{2}$ H). Mean absolute error (MAE) analysis confirmed that 23  $\delta^{18}$ O remained stable under all tested conditions (<0.6%), whereas  $\delta^2$ H was more 24 sensitive to storage duration, temperature, and flow rate. Optimal results were obtained 25 using 250 ml glass bottles, flow rates of 100-125 ml/min, and storage times of ≤24 26 hours under ambient conditions (20–25 °C), achieving MAE values of ±1.5% for δ<sup>2</sup>H. 27 Prolonged storage (>24 hours) increased isotopic variability, particularly for  $\delta^2 H$ . 28 Although vapor sampling cannot match the analytical precision of conventional liquid-29 water methods, it offers a practical, inexpensive, and non-destructive alternative for 30 isotope-based ecohydrological research. The validated protocol presented here 31 enables reliable vapor isotope measurements in both laboratory and field settings and 32 is especially advantageous in remote areas or locations with limited infrastructure. This 33 optimized method provides an accessible and robust tool for investigating plant water 34 uptake and soil-atmosphere interactions.

### Keywords

35

Water vapor, stable isotopes of water, isotopes  $\delta^{18}$  O,  $\delta^{2}$  H, storage time, storage temperature, flow rate, tightness, container

65

66 67

### 1 Introduction

The water stable isotopes oxygen-18 ( $\delta^{18}$ O) and deuterium ( $\delta^{2}$ H) are crucial for 39 40 studying the complex ecohydrological processes in different environments (Zhang et 41 al., 2022), as they allow the tracing of water through precipitation, soil, plants, surface 42 water and groundwater, thus allowing the analysis of variations in response to climatic 43 events (Tetzlaff et al., 2023). The use of isotopes has been fundamental to advances 44 in modern ecohydrology (Sprenger et al., 2016). Therefore, they have been used as 45 tools to understand hydrological fluxes in ecological and agricultural systems 46 (Haberstroh et al., 2024), especially in the context of climate change, water stress and 47 studies of exchanges between water, plants and the atmosphere (Dubbert and Werner, 48 2019; Allen and Kirchner, 2022). In addition, isotopes allow the tracing of water sources 49 and their redistribution in ecosystems Penna et al., (2018) and provide unique 50 information on transpiration, evaporation and water use strategies of vegetation 51 (Liebhard., 2022).

Isotope analysis has become an important tool in ecohydrology as it is able to directly 53 identify water sources and track water movements along the soil-plant-atmosphere 54 continuum (Beyer et al., 2020; Rothfuss et al., 2021). However, there is aneed to 55 generate continuous data sets, as high-resolution time series are essential to capture 56 the natural dynamics of ecohydrological and physiological processes (Simonin et al., 2013; Gaj et al., 2016). The methods used so far provide only incomplete information 57 58 and often rely on destructive sampling of soil and plant xylem, which requires the 59 extraction of material from forests and the combination of different techniques such as cryogenic vacuum extraction. This technique has been questioned due to its impact on 60 effects on the isotope composition, particularly for deuterium (Chen et al., 2020; Allen 61 62 and Kirchner, 2022; De Deurwaerder et al., 2020; Wen et al., 2022). Despite these 63 limitations, it continues to be widely used in studies to monitor hydrological processes 64 in ecosystems (Barbeta et al., 2022; Koeniger et al., 2011).

To obtain continuous measurements, the scientific community has relied on in situ instruments such as Cavity Ring-Down Spectroscopy (CRDS), which enable direct measurement of stable isotopes in soils and plants under real field conditions (Kühnhammer et al., 2022). Although highly valuable for ecohydrological research, this approach involves logistical, technical, and financial limitations that constrain its deployment in challenging environments (Rothfuss and Javaux, 2017; Volkmann et al., 2016). Moreover, the desired spatiotemporal resolution for isotopic analysis of water is not always achievable, as many techniques remain confined to the laboratory or are impractical in certain settings. In this context, field-equilibrated water vapor sampling emerges as a crucial alternative for generating reliable isotopic information, with broader applicability across natural, agricultural, and experimental environments, and without the need for destructive extractions (Gralher et al., 2021; Kübert et al., 2020; Galewsky et al., 2016). In practical terms, water vapor sampling is more flexible, more cost-effective, and better suited to remote sites; it allows repeated sampling of soils

98

and trees at the desired frequency without destructive extraction, avoids liquid water extraction in the laboratory, and does not require an isotope analyzer in the field.

81 In this way, recent studies have explored innovative methods for water vapor sampling, 82 developing strategies that allow the collection and storage of water vapor without 83 significant changes in its isotopic signature (Diekmann et al., 2025). In this context, 84 Magh et al., (2022) developed the Vapor Storage Vial System (VSVS), which does not rely on direct field measurements but allows the equilibration, collection and storage 85 of water vapor in vials for subsequent laboratory analysis. However, they reported 86 87 variations of 0.6 to 4.4 % for  $\delta^2$ H and 0.6 to 0.8 % for  $\delta^{18}$ O after a storage period. In 88 addition, Havranek et al., (2023) implemented and evaluated an automated method, 89 the Soil Water Isotope Storage System (SWISS), which combines several components such as permeable probes, glass flasks, stainless steel tubes and switching valves. 90 91 This system allows automated sampling and storage of equilibrated vapor for later 92 laboratory analysis. The authors reported a precision of  $\pm 0.9$  % for  $\delta^{18}O$  and  $\pm 3.7$  % 93 for  $\delta^2$ H after up to 14 days of storage.

Innovations in water vapor sampling and storage methods have reduced spatial and temporal demands, enabling approaches that are more practical, easier to implement, more reproducible, and that improve data reliability. These advances help narrow existing uncertainty gaps and provide a robust, complementary alternative to established methods, including direct measurement. In this context, Herbstritt et al., (2023) developed a sampling technique based on inflatable bags with diffusion-tight seals and validated it with results showing variations of 0.4 % for  $\delta^{18}$ O and 1.9 % for  $\delta^{2}$ H, which was a significant improvement compared to previous studies. However, methodological development has continued with new alternatives and increasing complexity. In this sense, Dahlmann et al., (2025) implemented a system for water vapor extraction using permeable membranes and special gas bags (Multi-Layer Foil Bags with stainless steel fitting, Sense Trading B.V., Netherlands) for storage. They also tested the possibility of reusing containers and reported relatively small variations after 24 hours of storage, with values between 0.7 and  $\pm 2.3$  % for  $\delta^{2}$ H and 0.2 to 0.9 % for  $\delta^{18}$ O.

Although these studies represent important advances, none of them provides a technical, practical and comprehensive protocol with optimal sampling and 110 measurement conditions. With our work, we attempt to overcome this challenge by 111 112 identifying the most suitable vessel among several options investigated and 113 establishing parameters such as flow rate, temperature and storage time that ensure 114 isotopic stability. Nevertheless, some research questions remain unanswered: How 115 can practical, reliable, cost-effective and repeatable water vapor protocols be 116 developed for different natural environments; How do extreme conditions affect the 117 isotopic signal, especially when no defined measurement time, temperature control or 118 suitable vessel is available; and to what extent can water vapor sampling generate 119 continuous data sets that capture ecohydrological dynamics?

- To overcome these limitations, we propose a practical, reliable and minimal-invasive
- method for water vapor extraction, developed and validated under controlled laboratory
- conditions before it can be applied in the natural environment. The method consists of
- collecting water vapor in a container under defined flow rates and storage temperature
- conditions and then analyzing it in the laboratory using laser spectroscopy (CRDS).
- With this approach, we aim to achieve precise results within defined time periods while
- enabling the creation of continuous and comparable data sets across time and space.
- The objectives of this research are clearly and concisely stated, emphasizing the
- importance of the practicality, non-invasiveness and efficiency of the proposed
- method, as well as its potential application both in the laboratory and in the field:
- Develop and validate a practical, minimally invasive, and efficient method for collecting water vapor samples for application in various environments.
  - Conduct laboratory tests to evaluate the effects of storage time, temperature, and container type on the stability and precision of the method.
- and container type on the stability and precision of the ldentify the optimal conditions for water vapor storage.
- Innovation. The novelty of this method lies not only in proposing practical and low-cost
- solutions for collecting and storing water vapor in the laboratory, but also in the
- integrated evaluation of factors that directly affect its isotopic composition. Rather than
- analyzing isolated effects, we adopt a comprehensive multifactor approach that
- simultaneously considers the type of container, sampling flow rate, temperature, and
- storage duration. This framework identifies operational conditions and decision criteria,
- facilitates application in natural environments, and maximizes data quality after
- laboratory analysis.

### 143 2 Materials and methods

- This section describes the experimental procedures and the tests carried out. The
- detailed list of materials, instruments and equipment used in the experiments can be
- found in S1.

132

### 147 2.1 Formal testing of sampling materials and methods

### 148 2.1.1 Overview of the systematic experimental testing

- With the aim of validating a new semi-in-situ method for measuring stable water
- isotopes from water vapor samples by experimental tests, four main experiments were
- performed in the laboratory. These are detailed in Fig. 1.

153154

155156

157158

159160

162163

167

**Figure 1.** Experimental scheme for the evaluation of the water vapor storage containers and optimization of the sampling method. The blue box represents the tests performed to evaluate the tightness and isotopic stability of the containers. The boxes connected by green arrows illustrate the decision-making process to determine the most suitable container. The box with the green lines represents the optimization of the sampling method performed with the most optimal container. Some parameters such as syringe sizes, flow rates, and isotope references were tested.

We carried out a sequence of experiments in order to develop the most suitable sampling protocol. First and foremost, it was crucial to test different storage containers (Tests 1 and 2, see Fig. 1) under different storage conditions and temperatures. Based on this first stage (subsequently referred to as *storage container testing*, see Fig. 1), the best-performing sampling container was identified. The subsequent testing was designed in order to find the ideal sampling setup and parameters (Tests 3 and 4, Fig. 1) and was performed only using the most suitable storage containers. This second stage of testing is subsequently referred to as *sampling procedure testing* (see Fig. 1).

The parameters under which the experiments were conducted were precisely defined. These included specific procedures for evaluating the most important variables such as storage time, storage temperature and flow rate.

### 2.1.2 Storage container testing

- The first phase of the experiments focused on evaluating different storage containers
- to assess their tightness and their ability to maintain the isotopic stability of water vapor.
- To this end, two tests were carried out: one with dry air and the other with a reference
- with known isotope values.

190

195

206

207

208

- The dry air and water vapor samples were subjected to three temperature conditions.
- Two of these were controlled and maintained constant temperatures: a Fridge at 4°C
- and an Oven at 40°C. The third condition, in the Room, where the temperature
- fluctuated between 20 and 25 °C, was uncontrolled as it corresponded to the normal
- ambient temperature of the laboratory. Three storage times were tested for each
- temperature condition: 6, 24 and 72 hours (0, 1 and 3 days).
- The following experiments were conducted in this phase:
  - The tightness of different types of containers was evaluated under different temperature conditions and storage times with the objective to determine the container with the least increase in H<sub>2</sub>O concentration due to intrusion of atmospheric air.
  - 2. The same containers and conditions as in the first experiment, but with water vapor samples of a known isotopic reference. The stability of the isotopic composition was evaluated to determine the container with the smallest deviation in isotopic concentration with respect to the reference or target value.
- These tests were essential for the selection of the optimal container. Decision criteria were defined for this purpose (Figure 1). These criteria enabled an objective evaluation of the containers and thus ensured the isotopic stability of the vapor samples during transport and storage until the time of measurement.

### 2.1.3 Sampling procedure testing

- The second phase of the experiment was carried out with the container that proved to 196 197 be the most suitable. The main objective was to optimize the process of water vapor sampling and to determine the optimal conditions for the method developed in this 198 199 study. In addition, this experimental phase included a longer storage period, with 200 samples being kept for up to 7 days (168 hours) after the first day of sampling. This 201 allowed a more accurate assessment of whether the isotopic composition of the vapor 202 exhibited the same variation observed after 3 days or whether it increased. In other 203 words. This phase aimed to confirm the importance of avoiding too long intervals 204 between sampling and measurement of the samples.
- Summary of the experiments carried out:
  - **3.** The tightness test was repeated only with the container selected as the most optimal container. Dry air was used and the effect of syringe size on the sampling results was evaluated.
- 4. The optimum flow rate for the water vapor sample was determined. Three isotopic references were used to evaluate the isotopic behavior under different temperature conditions and storage times.
- In test 3, which was conducted with dry air and using two syringe sizes, the samples were exposed to three temperature conditions: 40 °C, 20–25 °C and 4 °C (Oven, Room

- and Fridge) and four storage times: 6, 24, 72 and 168 hours (6 hours, 1, 3 and 7 days).
- In Test 4, the experimental conditions included four sampling flow rates (35, 75, 100
- and 125 ml/min) and three isotope references. The series started at 35 ml/min, which
- corresponds to the operating capacity of the CDRS isotope analyzer. For each
- temperature condition (Oven, Room, and Fridge), water vapor samples were collected
- from the three isotope references at the four flow rates and stored for 6 hours, 1, 3,
- and 7 days (see Fig. 1).
- The purpose of this phase was to determine the most reliable combination of syringe
- size and sampling flow rate that would ensure isotopic precision under the
- experimental conditions of temperature and storage time.

### 224 **2.1.4** Preparation of the containers prior to the experiments.

- Before we started the experiment, several preparatory measures were carried out. The
- 250 ml infusion glass bottles ND 32 were dried overnight in an oven at 60 °C to remove
- residues and then sealed with a butyl stopper and an aluminum cap secured with a 32
- 228 mm aluminum cap closure clip. The glass bottles were rinsed with dry air for 5 minutes
- before sampling. After measuring the samples, they were cleaned and dried again for
- 24 hours at 60-80 °C for reuse. For the 500 ml Aluminum-Zip bags with zippers, silicone
- was applied as a septum for the syringes and dried at Room temperature for three
- 232 days. They were then baked overnight at 50 °C to remove organic residues. The bags
- were tightly sealed by folding the top opening and securing with folding clips without
- using heat sealing to prevent the emission of volatile organic compounds. These bags
- were not reused. The 1 I FlexFoil sample bags were purged three times with dry air
- through the valves prior to sampling to minimize memory effects and allow reuse.

### 237 **2.1.5 Tightness of the containers**

- After the preparatory measures, the containers were filled with dry air from a cylinder
- regulated by a pressure regulator connected to an electronic mass flow controller. The
- air was passed through PTFE tubing (outer diameter: 1/8") connected to a special
- syringe depending on the phase of the procedure.
- Fine syringes (0.80 x 40 mm) were used for sampling to minimize the size of the
- perforations in the septa. Larger diameter syringes (1.20 x 80 mm) were used during
- the measurement as the thinner syringes did not allow sufficient flow, which could allow
- atmospheric air to enter and mix the sample.
- The 250 ml infusion glass bottles ND 32 were purged with the sample for 12 minutes
- at a flow rate of 100 ml/min, with an equilibrium between the inlet and outlet of dry air
- established by an additional connection to the CRDS isotope analyzer. An open outlet
- (open split) was used to avoid overpressure as the analyzer aspirates at 35 ml/min,
- which prevented damage.

- The 500 ml Aluminum-Zip bags were filled in six minutes under the same flow
- conditions by inserting the syringe directly into the silicone portion of the bag. For the
- 1 I FlexFoil® sample bags, a 1/4" PTFE tube was connected. This was connected via
- a Swagelok connector to a second 1/8" tube, which passed the dry air until the bag
- was filled.

### 2.1.6 Test of water vapor from an isotopic reference in the containers

- The same sampling configuration and container types were used as described above.
- In this experiment, the key difference was the use of saturated water vapor from a
- known isotopic reference to rinse the containers. Each container type (250 ml infusion
- glass bottles ND 32, 1 I FlexFoil sample bags and 500 ml Aluminum-Zip bags) was
- prepared in five replicates and stored for 6, 24 and 72 hours at different temperatures:
- Fridge, Room, and Oven.
- A first measurement was performed by direct sampling of the isotopic water vapor with
- the CRDS analyzer to compare the initial isotopic signal with that obtained after
- storage. The vapor flow was maintained at 100 ml/min for all containers. The sampling
- time was 12 minutes for glass bottles and six minutes for bags. The vapor was
- generated in a closed system (headspace) in which dry air flowed through a glass
- bottle containing the known isotopic reference. This system included connections with
- syringes and PTFE tubing regulated by an electronic mass flow controller (MFC) and
- with an open T-shaped split before entering the analyzer to prevent overpressure.
- The water vapor was fed into the containers using the same sampling principle as the
- dry air test. Only the samples from the glass bottles were measured in real time during
- rinsing; the bags, on the other hand, were sealed and measured later to compare the
- initial and final concentrations. The system configuration is shown in Figure 2.

**Figure 2.** Structure of the sampling system in the laboratory. A) Sampling of water vapor saturated with a known isotope concentration using 250 ml infusion glass bottles ND 32. B) Shows the filling of bags with saturated water vapor (500 ml Aluminum-Zip bags, and 1 I FlexFoil® sample bags).

### 2.1.7 Decision criteria for the selection of the optimal container

Seven evaluation criteria were defined, taking into account the factors relevant to the development of the study. These were divided into critical (criteria 1-3) and non-critical (criteria 4-6) categories.

The decisive criteria included: (1) isotopic stability, defined as the ability of the container to maintain the isotopic signature without significant changes; (2) tightness, determined as the efficiency of the closure system in preventing the exchange of dry air with the atmosphere; and (3) time to analysis, which was considered essential since the goal was to obtain data under conditions comparable to in situ measurements, with a maximum required interval of 0 to 1 day between sampling and isotopic concentration measurement. The non-critical criteria were: (4) reusability of the storage container, which was assessed as an aspect of sustainability and long-term cost reduction; (5) portability and transport, which refers to the weight and volume of the container and determines the logistical feasibility of transport; it was also assessed in terms of the

- risk of breakage, perforation or loss of tightness under transport conditions; and (6)
- cost, which was estimated based on the prices of suppliers in the European market.
- Each criterion was rated on a scale of 1 to 5 (1 = poor, 5 = excellent) and given a
- relative degree of importance: Isotope stability (50%), tightness (20%), time to analysis
- (10%), reusability (10%), portability and transport (5%), and cost (5%).
- The sum of the weighted scores enabled an objective comparison between the
- evaluated containers. The container with the highest total score resulting from the
- combination of decision criteria was considered the optimal container.

### 2.1.8 Evaluation of syringe size and sampling flow rate using the optimal container

### 2.1.8.1 Influence of syringe size on the tightness of the optimal container

Two different syringe sizes were evaluated to determine whether the size of the sampling opening affects tightness of the closure systems (e.g., the used septa) and if the used syringe is a constraint in terms of sampling velocity (flow rate). Tests were carried out with 250 ml glass bottles previously filled with dry air. After the samples had been taken with both syringes, the bottles were subjected to the conditions described in the test conditions section. This evaluation determined which of the syringe sizes provided the best performance in terms of maintaining the dry air sample at the same volume of H<sub>2</sub>O (ppm) sampled. This determined the most suitable syringe size for water vapor sampling.

# 314315316

303

### 2.1.8.2 Testing the flow rate of the sampling in the optimal container

- The precision of the water vapor sampling method was evaluated using a test with
- known isotope concentrations. Three reference standards were used: ALBI1 (enriched
- in deuterium oxide), BS (intermediate isotopic composition), and KEI (light isotopic
- composition).

Table 1. Liquid isotope concentrations of the reference standards.

| Standards | δ <sup>18</sup> Ο <sub>liquid</sub> [‰] | δ <sup>2</sup> H <sub>liquid</sub> [‰] | Isotopes     |
|-----------|-----------------------------------------|----------------------------------------|--------------|
| ALBI1     | -8.39                                   | 129.2                                  | Heavy        |
| BS        | -8.5                                    | -57.71                                 | Intermediate |
| KEI       | -21.1                                   | -158.08                                | Light        |

The water vapor samples were generated under controlled conditions in the laboratory

- and subjected to the test system described in the section on test conditions. Sampling
- was performed at four different flow rates: 35, 75, 100 and 125 ml/min to determine if
- the flow rate affected the isotopic signature of the samples. Each combination of

https://doi.org/10.5194/egusphere-2025-5295 Preprint. Discussion started: 21 November 2025 © Author(s) 2025. CC BY 4.0 License.

- standard and flow rate was repeated three times to ensure the stability of the results.
- The reliability of this test allows the uncertainty of the sampling method to be validated
- and the flow rate(s) to be identified that may have a significant impact on the accuracy
- of the isotope concentrations measured with water vapor.
- Before sampling, the 250 ml glass bottles were cleaned by placing them in an oven at
- 60-80 °C for 24 hours. They were then stored in a dry place (aluminum box with
- insulation) with a butyl stopper to prevent mixing with atmospheric moisture. They were
- also rinsed with a strong, dry stream of air for 1 to 2 minutes before use to ensure that
- the inside of the bottle was completely clean and dry. They were then sealed by placing
- an aluminum cap on the butyl stopper and pressing down with sealing pliers to ensure
- a secure and tight seal.
- During water vapor sampling, the temperature was monitored to ensure that it
- remained stable and recorded with a sensor (Sensirion AG, SHT4X SMART GADGET)
- that allowed a uniform temperature to be maintained throughout the experimental
- Room. Approximately 100 ml of the reference standard was placed in a 250 ml glass
- bottle to ensure headspace sampling. During the sampling time, we achieved a
- dynamic equilibrium and the isotope exchange reached a fractionation factor defined
- by the temperature.

344345

349350

355

359

The sampling system implemented makes it possible to inject the air stream passing through the bottle containing the reference standard to equalize and exchange the isotopes in the water vapor flushing the sampling bottles. A synthetic air cylinder was used, which had a pressure regulator with an approximate flow rate of 2 bar and was connected to an electronic mass flow controller. This was used to regulate or inject the air flow used for sampling. The connections were made with 1/8 PPT tubing. The end connected to the MFC had Swagelok connectors and the other had a Luer lock adapter with a large syringe inserted through the septum into the glass bottle. Then another shorter syringe was inserted to deliver the water vapor into the bottles. The three replicates were withdrawn simultaneously for 35 minutes, i.e. the water vapor flushed the three bottles at the same time, ensuring that all three bottles received the same amount of water vapor passed through tubing and syringes. The bottle with the standard injected the water vapor into the first replica, then into the second bottle, and finally into the third bottle, which was connected directly to the CRDS to measure the sample and obtain an initial reference. Before connecting to the Picarro, we performed an open split to avoid pressure on the devices. The configuration of the sampling system is shown in the diagram below.

**Figure 3**. Schematic of the water vapor sampling system with glass bottles and measurements with the CRDS isotope analyzer (Picarro L2130-i)

For each of the reference standards (3), samples were taken and subjected to the procedures described in the section on experimental conditions. For each combination of time, flow rate and standards, 36 samples were taken (4 combinations). A total of 144 samples were taken per temperature reservoir, giving a total number of 432 samples for the entire experiment.

### 2.1.8 Measurement of the water vapor samples

Before measurement, the water vapor samples were kept at Room temperature for one hour, making sure that this temperature was higher than the temperature measured during sampling. This was done to achieve a homogeneous temperature inside the bottles and to prevent condensation.

The measurements were carried out with a Picarro L2130-i isotope water analyzer, which measures the concentrations of  $\delta^{18}O$  and  $\delta^{2}H$  using Cavity Ring-Down Spectroscopy (CRDS). The samples in the 500 ml aluminum-Zip bags were connected directly to the isotope analyzer using a Swagelok connector and 1/8" diameter PTFE tubing. At the other end, a Luer lock adapter was attached to a syringe that was inserted through the silicone septum into the bag so that the isotope analyzer could directly extract the water vapor at a flow rate of approximately 35 ml/min. 1 l FlexFoil sample bags with a gas valve, the measurement was made through the valve connected to a 1/4" PTFE tube. This was connected via a Swagelok connector to a second 1/8" tube that led the vapor directly to the Picarro.

The 250 ml glass bottles were connected to the isotope analyzer via a PTFE tube attached to a Luer lock adapter with a 0.80x40 mm syringe inserted into the bottle. The other end of the tubing was connected via a Swagelok connector to a stainless-steel T-piece and from there to the isotope analyzer. In contrast to the bags, the bottles were connected to a second tube with a 1.20x40 mm syringe, which was used to introduce

a constant flow of dry air into the bottle at a rate of 40 ml/min, controlled by a mass flow controller (MFC). This strategy was used because this flow exceeds the demand of the isotope analyzer, which prevents the creation of a vacuum during the measurement and thus minimizes the risk of atmospheric air entering the sample.

By introducing dry air, the water vapor could flow through the connections directly to the isotope analyzer. The T-piece was used to avoid overpressure, ensure a continuous flow and prevent mixing with atmospheric air. An electronic rotameter was used to check that the excess air was vented through the T-piece (approximately 5-7 ml/min). This check ensured that the flow detected by the analyzer was solely from the sample and was not disturbed by atmospheric air.

Each sample was measured over a period of 6 minutes. Between each measurement, the isotope analyzer was purged with dry air for 2 minutes to clean the measurement system and minimize the risk of memory effects.

**Figure 4.** Setup of the measuring system with the Picarro L2130-i. A) Measurement of 250 ml infusion glass bottles ND 32. B) Measurement of bags (1 I FlexFoil sample bags and 500 ml aluminum-Zip).,

### 2.2 Processing of the data

The measurements were performed in the water vapor phase of the samples and the isotope ratios  $\delta^{18}O$  and  $\delta^{2}H$  as well as the water vapor concentration (H<sub>2</sub>O) in ppm were determined with the Cavity Ring-Down Spectroscopy (CRDS) isotope analyzer. These values are presented in delta notation ( $\delta$ ) relative to the international standard

- VSMOW (Vienna Standard Mean Ocean Water) according to equation 1 (Craig 1961
- and Coplen, 2011):

$$\delta sample = \left(\frac{R_{sample}}{R_{VSMOW}} - 1\right) * 1000 \left[\%\right] \tag{1}$$

- where: $R_{sample}$  is the isotopic ratio (<sup>18</sup> O/<sup>16</sup> O or <sup>2</sup> H/<sup>1</sup> H) of the sample, and  $R_{VSMOW}$  is
- the ratio of the international standard VSMOW.
- Subsequently, the first two minutes of each measurement were discarded due to
- possible memory effects in the system, as well as the last two minutes, as the water
- vapor concentration (ppm) was considerably low, which could influence the estimation
- of the isotopic composition. Therefore, data processing was based solely on the middle
- two minutes of each measurement.
- From these data, the average values per replicate were calculated and finally the
- isotopic values corresponding to the liquid phase of each sample were determined. For
- this purpose, the temperatures recorded during the sampling time were used and the
- equations for isotope fractionation in equilibrium between vapor and liquid were
- applied. These equations (2, 3 and 4) are based on the source temperature-dependent
- fractionation factor expressed in Kelvin [K] (Horita et al., 2008).
- Equation 2; equilibrium fractionation ratio (α) for  $\delta^{18}$ O

$$10^{3} \ln \alpha_{L/V} (^{18} \text{ O} / ^{16} \text{ O}) = -2.0667 - 0.4156 \left(\frac{10^{3}}{T}\right) + 1.137 \left(\frac{10^{6}}{T^{2}}\right)$$
 (2)

Equation 3; fractionation ratio at equilibrium ( $\alpha$ ) for  $\delta^2$  H

$$10^{3} \ln \alpha_{L/V} (^{2} H/^{1} H) = 52.612 - 76.248 \left(\frac{10^{3}}{T}\right) + 24.844 \left(\frac{10^{6}}{T^{2}}\right)$$
 (3)

Equation 4; liquid phase isotopic value

$$\delta_l = (\delta_n + 1000) * \alpha - 1000 \tag{4}$$

- where: $\delta_l$  is the liquid isotopic value,  $\delta_v$  is the vapor isotopic value, and  $\alpha$  is the
- fractionation factor for <sup>18</sup> O and <sup>2</sup> H obtained from equations 2 and 3.
- Using the fluid values obtained, a linear regression was calculated between the
- measured values and the known values. This regression was applied to correct and
- standardize all samples so that the final isotopic values could be corrected and
- determined in accordance with the international VSMOW standard.
- As part of the data processing, a quality check of the replicates was performed by
- comparison with the known isotopic signatures of the standards. To ensure the
- precision and reliability of the standardization, a quality filter was applied in Python,
- which used as an exclusion criterion those replicates that showed significant deviations
- from the expected value. For this purpose, all possible subsets of 2 and 3 replicates
- were evaluated for each combination of experimental conditions (standard type,
- temperature, storage time and flow rate), calculating for each subset the mean and

- standard deviation of  $\delta^{18}O$  and  $\delta^{2}H$  and their distance from the target value. The
- subgroup with the lowest value, defined as the sum of the standard deviations, was
- selected, favoring the subgroups with the highest number of replicates. This procedure
- made it possible to determine the most representative and reliable replicates for each
- experimental condition, thus maximizing both the accuracy and precision of the
- measurements.

### 452 2.3 Analysis of the data

- The data was analyzed with the Python programming language (3.13.1 version) using
- libraries such as Pandas, Numpy, Matplotlib, Seaborn and Scipy. Graphical and
- statistical comparisons were performed to evaluate the effects of storage temperature
- (Oven, Room and Fridge) and container type on the stability of isotope values. The
- differences between the groups were visualized using boxplots and statistically
- evaluated using non-parametric tests such as Kruskal-Wallis, as the data did not follow
- a normal distribution. Furthermore
- To validate the precision of the isotope results, the Mean Absolute Error (MAE)
- between the measured values and the known values of the reference standards was
- calculated. This metric was used to quantify the average degree of deviation from the
- reference value, which provides an objective measure of the quality of the fit.
- Equation 5; Mean Absolute Error (MAE)

$$MAE = \frac{1}{n} \sum_{i=1}^{n} |y_i - \hat{y_i}|$$
 (5)

- Where: n is the total number of observations,  $y_i$  is the observed (actual) value,  $\hat{y_i}$  is the
- predicted (estimated) value, and |. | is the absolute value.
- 3 Results

### 469 3.1 Storage container testing

### 470 3.1.1 Tightness of storage containers.

- Fig. 5 shows the results obtained for the tightness of the various containers. It shows
- a clear increase in the H<sub>2</sub>O concentration (ppm) in the dry air samples depending on
- the type of container, storage time, temperature and volume.
- The 250 ml infusion glass bottles with ND 32 showed a significant increase in H<sub>2</sub>O
- concentration after 3 days (72 h) of storage, reaching ~2200 ppm at 40 °C and ~1250
- 476 ppm under ambient conditions (20–25 °C), while values at 4 °C remained lower at ~480
- 477 ppm. The 500 ml aluminum-Zip bags showed a similar pattern, but with the strongest
- increase under ambient conditions (~1750 ppm), while under the extreme temperature
- conditions, i.e. Oven (40 °C) and Fridge (4 °C), they showed comparable

concentrations of ~1000 ppm after 3 days. In contrast, the 1 I FlexFoil sample bags 481 showed the least variability. They maintained values between ~100 and ~600 ppm 482 across all storage conditions and times with no significant increase.

**Figure 5.** The graph shows the changes in water vapor concentration ( $H_2O$ , in ppm) in dry air samples after different storage times (0, 1 and 3 days) in three types of containers (250 ml infusion glass bottles with ND 32, 1 l FlexFoil sample bags and 500 ml aluminum-Zip bags) and under three temperature conditions (Fridge, Room and Oven).

### 3.1.2 The efficiency of containers in preserving the isotopic composition of water vapor.

An isotopic reference was used to evaluate whether the isotopic composition remained stable after sampling in three types of containers and storage at different temperatures and times.

In addition to the results presented in this study, further tests were carried out with 100 ml infusion glass bottles ND20 and 1 I QEC Tedlar® bags. However, the results obtained with these containers were inconsistent; therefore, these data are not reported and were excluded from the main analysis.

The results show a clear isotopic deviation of the measured values from the known values of the isotopic reference source. This deviation strongly depends on temperature, storage time and container type and increases with increasing storage time of the sample. In the 250 ml infusion glass bottles ND 32, the isotopic deviations in  $\delta^{18}$ O remained stable within a range of  $\pm 0.9\%$  over all three storage times (0, 1 and 3 days) under all temperature conditions. Slight variations in  $\delta^{2}$ H were observed, especially for samples stored in the Oven (+5%) and Fridge (-5%) after three days, while samples stored at Room temperature showed a stable deviation of  $\pm 0.5\%$  at all storage times. Overall, the results indicate that the analysis of the samples between day 0 and 1 provides reliable values without deviations of more than  $\pm 1\%$  (Fig. 6).

In contrast, the 1 I FlexFoil sample bags exhibited δ<sup>18</sup>O deviations within ±1‰ for samples stored under Fridge and Oven conditions during days 0, 1 and 3. However, samples stored at Room temperature showed deviations of –1.35‰ from the first day

and a +1.08% enrichment after three days. The deviations in  $\delta^2 H$  were more pronounced from 0 to 1 day, with deviations for Room at -3%, and values reaching up to ±8% after three days under Room and Fridge conditions. In contrast, in the Oven, deviations remained closer to +2‰, indicating a clear trend toward isotopic enrichment under these conditions. These results show that deviations are expected in this type of container from the first day of storage and tend to increase over time. Additionally, the fact that the deviations varied depending on the temperature conditions indicates that this material reacts differently to temperature changes, which significantly affects the isotopic signature of the water vapor (Fig. 6).

Finally, the 500 ml aluminum-Zip bags showed the largest increase in isotopic deviations for  $\delta^{18}$ O, reaching -3% under Fridge and Room conditions. In the Oven, the samples showed values of -2.5% on days 0 and 1, followed by a slight enrichment to +1% on the third day. For  $\delta^2$ H, isotopic fractionation towards depletion was observed under all three temperature conditions, with deviations of up to -25% in the Room, -16% in the Fridge and -12% in the Oven. Taken together, these results indicate that the aluminum-Zip bags preserve the isotopic composition of the water vapor the least reliably (Fig. 6).

**Figure 6.** Isotopic deviations ( $\delta^{18}O$  and  $\delta^{2}H$ ) in water vapor samples stored in different containers (250 ml infusion glass bottles ND 32, 1 l FlexFoil sample bags and 500 ml aluminum-Zip bags) at three storage temperatures (Fridge, Room, Oven) for 0, 1 and 3 days. The dashed line represents the reference value (0‰), which shows no isotope deviation

### 3.1.3 Selection of the optimal container

The 250 ml glass bottles ND 32 achieved the highest overall score of 4.25 (85%) and were therefore the optimal container for water vapor sampling. They were

- characterized by their isotopic stability during the first days of storage (±1% deviation
- for both  $\delta^{18}O$  and  $\delta^{2}H$ ; Fig. 6). Although they showed a greater increase in  $H_{2}O$
- compared to the other containers (Fig. 5), their reusability, low cost on the European
- market and reasonable transport stability making them a practical and cost-effective
- option for field sampling campaigns.
- As a second option, the 1 I FlexFoil® sample bag performed well in terms of tightness
- with a score of 3.70 (74%) (Fig. 5), although it exhibited greater isotopic deviations
- than the glass bottles, especially under Room temperature conditions (Fig. 6). Its
- reusability is limited (maximum three cycles) and it is the most expensive container on
- the European market. Its main disadvantage lies in its portability, as the occupied
- volume increases when filled, making it difficult to transport and store a large number
- of samples and increasing the risk of damage during handling.
- Finally, the 500 ml aluminum bags received the lowest score of 1.95 (39%). This poor
- performance was mainly due to their limited isotopic stability (Fig. 6) and the increase
- in H<sub>2</sub>O observed during the tightness tests, especially after 3 days of storage. Although
- they have the advantage of low cost, their reusability is practically limited to a single
- use. In addition, their portability and resistance to transportation are unfavorable, as
- the occupied volume makes it difficult to handle multiple samples and the material is
- prone to punctures.
- Overall, the glass bottles have advantages that establish them as the optimal container
- for water vapor samples and the measurement of  $\delta^{18}$ O and  $\delta^{2}$ H isotopes. The criteria
- in favor of these choices relate in particular to their ability to maintain a reliable isotopic
- signature during the first day of storage under different temperature conditions. Figure
- 7 shows the ratings for each decision criterion and the resulting final score. Additional
- information on the individual and the rationale for each assessed criterion can be found
- in Table S2.

**Figure 7**. Evaluation and scoring of decision criteria (1 = poor; 5 = excellent) for the optimal container for water vapor isotope measurements. Critical criteria: isotopic stability (50%), tightness (20%) and time to analysis (10%). Non-critical: reusability (10%), portability and transport (5%) and cost (5%).

### 3.2 Results of syringe size and sampling flow rate in 250 ml infusion glass bottles (ND 32)

### 3.2.1 Effect of syringe size on the tightness of 250 ml infusion glass bottles

The results of the evaluation of the tightness of 250 ml glass bottles with synthetic dry air and the effect of the size and diameter of the syringes identified as small and big (0.80x40 mm and 0.80x80 mm and 1.20x40 mm and 1.20x80 mm) during sampling.

It should be noted that the values obtained include the initial concentration of  $H_2O$  between 200 and 350 ppm. The results show that the  $H_2O$  content in ppm increases progressively with storage time, being higher at higher temperatures. Samples stored at 40 °C reached the highest concentrations after 7 days, around ~4000 ppm, followed by samples stored at 20-25 °C at ~2000 ppm, and finally samples stored at 4 °C, which had the lowest  $H_2O$  concentrations of ~700 ppm after 7 days.

The trends in  $H_2O$  concentration in the dry air samples in the 250 ml glass bottles were consistent between the two syringe sizes, indicating that syringe size had no significant effect on the increase in  $H_2O$  concentration as a function of storage time and temperature, as shown in Figure 8.

**Figure 8.** Absolute  $H_2O$  concentration (ppm) in synthetic dry air samples stored in 250 ml glass bottles and analyzed with small and large syringes at different temperatures and times. The initial value (0 hours) of  $H_2O$  concentration was approximately 200–300 ppm; the values obtained correspond to the total measured (including the increase during storage).

### 3.2.2 Effect of sampling flow rate on water vapor stored in 250 ml infusion glass bottles (ND 32)

Water vapor sampling was conducted at four flow rates (35, 75, 100, and 125 ml/min). The  $H_2O$  concentration reached a plateau at approximately 25 minutes for the 100 and 125 ml/min runs; at 75 ml/min, a plateau was observed around 31 minutes, while at 35 ml/min, the concentration (ppm) did not fully stabilize within 35 minutes. A stable plateau was observed at 100 and 125 ml/min, but at 35 and 75 ml/min, the isotopic plateau developed more gradually over the sampling period, particularly for  $\delta^2H$  (Fig. S4).

### 3.2.2.1 Impact of sampling flow rate on $\delta^{18}$ O isotopic composition

The results obtained for  $\delta^{18}O$  show that the four flow rates used for water vapor sampling generally have deviations close to 0% and a low dispersion or variability of less than  $\pm 1\%$ . However, the results show that the best isotope values are obtained at flow rates of 75 to 125 ml/min. The same behavior is observed for storage times. However, the samples measured after 6 and 24 hours of storage show the best isotopic stability, with a mean value very close to 0% and less scatter. From 72 to 168 hours of storage, the samples show a slight increase in deviations, although they remain good.

The temperature conditions under which the samples were stored differed slightly. Under cold conditions (Fridge) they showed the least scatter and deviation, with a median of 0.03‰, indicating good stability. Stable results were also obtained in the

Oven, but with a larger scatter, with an IQR of 0.79‰. Meanwhile, the environmental conditions (Room) showed deviations closest to 0 at 0.01‰, which is the best isotopic stability. However, the samples from the Fridge seem to have the greatest stability at first glance, but they are the ones with outliers of -2.5‰. Finally, the KEI and ALBI1 standards have the greatest stability and consistency, with deviations close to 0‰ and less scatter. In contrast, BS shows higher deviations and scatter compared to the other standards. It can also be clearly seen that 75% of the deviations of the measured values from the target values for  $\delta^{18}$ O are below 0.5‰ and 25% with values between -0.3 and -0.5‰. There are outliers in the deviations of up to -2.5‰, especially with a flow of 35 ml/min, a storage time of 168 hours and samples stored in the Fridge. Therefore, storage of the samples for less than 24 hours is optimal for maintaining the isotopic stability of  $\delta^{18}$ O (Figure 9).

**Figure 9**. Isotopic deviations in  $\delta^{18}O$  ( $\Delta\,\delta^{18}O$ , ‰) of water vapor samples as a function of four factors: sampling flow rates (ml/min), storage time (hours), storage temperature (Fridge, Oven and Room) and standard type (ALBI1, BS and KEI). The center line of the box represents the median, the edges of the box represent the range of quartiles (Q1-Q3) and the shiskers are the lines that extend from the edges of the box and exclude the outliers (circles).

657

658 659

669

### 3.2.2.2 Impact of sampling flow rate on $δ^2H$ isotopic composition

The deviations in  $\delta^2$ H of the water vapor samples varied depending on the flow rate, time and storage conditions as well as the standard used. The flow rate of 35 ml/min 632 633 gave the greatest variability in the results with an IQR (interquartile range) of ±5.38% 634 and the deviation from the median was higher than the other flow rates at ±0.89‰, 635 indicating less stability when using this flow rate for water vapor sampling. In addition, outliers of up to ±15% were detected. Meanwhile, the flow rate of 75 ml/min showed 636 637 the lowest deviation in the median at ±0.18‰, indicating good precision with respect to the target value, but the IQR showed a dispersion of ±2.49% with outliers of ±5%. 638 The rate of 100 ml/min showed a deviation from the median of ±0.32%, higher than 639 the flow rate of 75 ml/min, but similarly variable with a dispersion of ±2.61%. Finally, 640 641 the flow rate of 125 ml/min showed a deviation from the median of ±0.28%, indicating 642 stability and consistency of the sample values with respect to the target values, with the IQR indicating a dispersion of 1.86%. It can be observed that when using flow rates 643 644 of 75 to 125 ml/min, 75% of the values have deviations of more or less below ±1.5% 645 and the variability is more homogeneous and less scattered. 25% have deviations of -646  $\pm 0.5\%$ , but there are outliers between  $\pm 5\%$  and  $\pm 4\%$ .

In terms of storage time, the deviations tended to increase slightly over longer periods. 648 After 6 hours, lower medians were found with ±0.18% and an IQR of ±2.62%, indicating 649 greater precision and moderate consistency of the samples. After 24 hours, the median 650 remained close to zero at ±0.22‰, with the lowest scatter observed at an IQR of 651 ±2.04‰, indicating an optimal time point for isotopic stability. After 72 hours, the median increased to a deviation of ±0.74‰, with a higher IQR of ±3.48‰, indicating 652 653 greater deviation and variability. Finally, the samples stored for 168 hours reached a median deviation of ±0.76‰, the highest dispersion of ±3.76‰ and outliers of ±15‰, 654 indicating lower isotopic stability. 655

The samples stored at Room temperature had the lowest median of ±0.14‰ and the lowest dispersion according to the IQR of ±2.72‰, indicating greater precision and stability. The oven-stored samples had a median of ±0.26‰ with a scatter of ±2.82‰ according to the IQR, indicating good stability but slightly greater scatter than the Room samples. Meanwhile, samples stored at cold temperatures (Fridge) showed a higher median of ±0.59‰ and the highest IQR scatter of ±3.38‰, indicating less stability under these conditions with outliers of ±15‰. Finally, the standards ALBI1 and KEI had the lowest deviation of -0.08 and -0.15 ‰ respectively, however ALBI1 had a larger spread with an IQR of ±3.16‰ with outliers of ±15‰, while KEI had a lower spread (IQR of ±1.83‰). BS happened to show the highest deviation values with ±2.03‰ and an IQR of ±3.68‰, indicating a higher dispersion and lower stability compared to the other standards (Figure 10). We can see that we have isotopic deviations with outliers for all factors, but they are more noticeable when we use a flow rate of 75 ml/min, a storage time of 168 hours, a temperature in the Fridge and the standards ALBI1 and BS (heavy and intermediate)

**Figure 10.** Isotopic deviations in  $\delta^2 H$  ( $\Delta \delta^2 H$ , ‰) of water vapor samples as a function of four factors: sampling flow rate (ml/min), storage time (hours), storage temperature (Fridge, Oven and Room) and standard type (ALBI1, BS and KEI). The center line of the box represents the median, the edges of the box represent the range of quartiles (Q1-Q3) and the shiskers are the lines that extend from the edges of the box and exclude the outliers (circles).

### 3.2.3 Optimal conditions for water vapor sampling in 250 ml infusion glass bottles (ND 32)

The corresponding analysis of the results suggests that the greatest consistency and precision of the isotope values obtained in the water vapor measurements was achieved when sampling was performed at a flow rate of 125 ml/min. However, the values become consistent as the flow rate increases from 75 ml/min. For storage times between 6 and 24 hours and at ambient temperatures (20-25 °C), the distribution of values is in narrower fields, with low IQRs and few outliers, reflecting the homogeneity and consistency of the measurements. Therefore, the above conditions with low IQRs, medians close to zero and few outliers are best suited to obtain reliable and repeatable results for the  $\delta^{18}\text{O}$  and  $\delta^{2}\text{H}$  isotope concentrations of water vapor, as shown in Figure 11.

**Figure 11.** Optimal experimental factors and conditions to minimize deviations in water vapor isotope measurements. Panel A shows the deviations for  $\delta^{18}O$  and panel B shows the deviations for  $\delta^{2}H$ .

## 3.2.4 Mean Absolute Error (MAE) of the results obtained under optimal sampling conditions

The Mean Absolute Error (MAE) analysis confirmed that the experimental conditions have a significant impact on the precision of the water vapor isotope measurements, especially for  $\delta^2H$ . In particular, it was found that the best deviations between

measured and target values were obtained when flow rates of 75, 100 or 125 ml/min were used for water vapor sampling and samples were stored between 6 and 24 hours at ambient conditions with temperatures between 20 and 25 °C. This means that the best isotope values can be obtained with the combination of these conditions, with deviations of  $\pm 1.52$  to  $\pm 1.89\%$ . However, at a flow rate of 35 ml/min, the deviations increase to 3.55%, and if the samples are stored for longer than 24 hours (e.g. 72 and 168 hours), the deviations also increase. Similarly, very low (4 °C) or extreme (40 °C) temperatures lead to larger deviations. For  $\delta^{18}$ O, the MAE shows that the deviations are small, with a precision of  $\pm 0.6\%$  under all evaluated conditions, indicating that only  $\delta^{2}$ H is directly affected by variations in temperature, air flow, storage time and isotope concentration (heavy, medium and light), as shown in Figure 12.

**Figure 12.** Mean Absolute Error (MAE) for  $\delta^{18}O$  (A) and  $\delta^{2}H$  as a function of flow rate, storage time, temperature and standard type. The bars highlighted in black mark the deviations with the lowest MAE value and the optimal conditions for  $\delta^{2}H$  in field B. No bars are highlighted for  $\delta^{18}O$ , as all factors have deviations of less than  $\pm 0.6\%$ .

The analysis of the optimal conditions for water vapor extraction shows that more than one flow rate may be suitable depending on the measurement duration in combination with the temperature conditions. For samples stored for 6 hours, both 100 and 125 ml/min showed small and consistent errors at  $\delta^2 H$  ( $<\pm1.3\%$ ) and  $\delta^{18}O$  ( $<\pm0.5\%$ ), while 75 ml/min was acceptable for  $\delta^{18}O$  but had a variability of up to  $\pm3\%$  at  $\delta^2 H$  and therefore could only be considered as a partial alternative. After 24 hours, the flow rate of 125 ml/min proved to be the most optimal as it gave errors below  $\pm0.5\%$  for  $\delta^{18}O$  and  $\pm0.3-1.5\%$  for  $\delta^2 H$ . However, 100 ml/min could still be considered acceptable under certain conditions, provided that the vapor samples are not exposed to extreme temperatures, as this would otherwise lead to  $\delta^2 H$  errors of over  $\pm2\%$ . Finally, during prolonged storage (72 and 168 hours), only the flow rate of 125 ml/min remained within the range of  $\pm1.5$  to  $\pm4\%$ , while the other flow rates exhibited greater variability, resulting in lower reliability. Therefore, if the samples are to be analyzed within a short time (6 h), 100 ml/min is the optimal flow rate, while 125 ml/min ensures greater stability under storage conditions of one day or longer (>24 h) (see Table 2).

**Table 2**. Recommended optimum flow rates for water vapor samples as a function of time and storage conditions in 250 ml glass bottles (ND 32).

| Storage time                | Flow rate<br>(ml/min)                      | Optimal conditions      | Expected MAE δ <sup>2</sup> H (‰) | Expected MAE δ <sup>18</sup> O (‰) |
|-----------------------------|--------------------------------------------|-------------------------|-----------------------------------|------------------------------------|
| ≤ 6 h                       | 100, 125 and<br>(75 alternative<br>option) | 4 °C<br>and<br>20-25 °C | < 1.3<br>(75 ml/min to<br>3‰)     | <0.5                               |
| 24 h (1 day)                | 125 (optimal)<br>and<br>100(acceptable)    | 20-25 °C<br>y<br>40 °C  | 0.3 – 1.5                         | 0.2 – 0.5                          |
| >24 h (72 and<br>168 hours) | 125                                        | 20-25 o 40<br>°C        | >1.5                              | 0.4 – 0.6                          |

The comparison of the MAE between the isotopes shows that  $\delta^2H$  was considerably more sensitive and exhibited variations in the mean absolute error across the flow rates and storage times. In particular,  $\delta^2H$  highlighted the importance of correctly selecting the optimal flow rate for sampling, especially when deciding on the measurement time ( $\leq 6$  hours or  $\geq 1$  day) and the temperature conditions under which the samples should be stored. In contrast,  $\delta^{18}O$  showed stable values, low errors (

745746

748749

751

752753

754755

757

758759

761

762763

765766

768

**Figure 13.** summary of the mean absolute error (MAE) for  $\delta^2 H$  and  $\delta^{18} O$  as a function of sampling flow rate and storage time. Each bar represents the average MAE for a given flow rate (75, 100 or 125 ml/min) and storage time (6 h, 24 h, >24 h), calculated from all combinations of temperature (Fridge, Room, Oven) and isotope references (ALBI1, BS, KEI). The error bars correspond to the standard deviation between these combinations and reflect the variability associated with the storage conditions and isotope type.

#### 4 Discussion

### 4.1 Evaluation of containers and criteria for water vapor storage

Our results show that vapor exchange of the sampling container with dry air depends directly on the closure system and is amplified by time and temperature. In 250 ml glass bottles, the H<sub>2</sub>O concentration increased from 480 ppm at 4 °C to 1250 ppm at 20-25 °C and 2200 ppm at 40 °C over 72 hours, consistently and systematically. This indicates that for the 250 ml infusion bottles, the crimp-cap closure with a silicone septum is the critical pathway for diffusive exchange caused by syringe puncturing of the septum during sampling. In 500 ml aluminum-zip bags, a similar pattern was observed, with the largest increase under ambient conditions (up to 1750 ppm) and comparable values of 1000 ppm at 4 °C and 40 °C, suggesting that the zip-type seal allows slight exchange via microleaks. In contrast, 1 I FlexFoil® bags showed lower variability, with values between 100 and 600 ppm and no significant increase in the water vapor concentration over time and with temperature. Comparing our results with those reported by Magh et al. (2022), who evaluated a Vapor Sample Vial Storage System (VSVS) using a dry air diffusion test during storage in 50 ml crimp-neck vials sealed with double-layer PTFE/butyl caps, they recorded increases from 600 ppm to 1300 ppm after 14 days, despite the use of high-quality materials.

Although glass is a stable material, the observed increase in the 250 ml infusion bottles does not depend on container volume but on the quality of the closure and seal. This is demonstrated by the valve system of the FlexFoil® bags, which showed the greatest stability, with no significant increases. In contrast, the material composition and/or the zip-type sealing mechanism of the aluminum bags exhibits a non-linear response to thermal conditions, suggesting that material permeability may be temperature dependent and display extrema under thermal extremes. This differs from the pattern observed in glass bottles, where a pronounced thermal dependence of diffusion or permeability associated with septum puncturing is apparent, with increases as temperature rises (4 °C 

837

845

847

single-use; however, Herbstritt et al. (2023) recommend preconditioning them with repeated fillings of isotopically homogeneous vapor to reduce memory effects and potentially improve measurement precision. However, the procedure is laborious and 812 must be repeated to be effective (Herbstritt et al. 2023). The one-liter FlexFoil bags 813 were more expensive and reusable but require a rigorous flushing procedure, as 814 815 described by Dahlmann et al. (2025). Overall, glass bottles offer the best balance 816 between price in the European context and high reusability (Table S1). Our most 817 important criterion was the preservation of the vapor's isotopic signature. In this 818 context, 250 ml glass bottles achieved the highest score (85%) among all evaluated 819 containers, making them the optimal choice for water vapor sampling. Their isotopic 820 stability during the first days was  $\pm 0.9\%$  for  $\delta^{18}O$  and  $\pm 1\%$  for  $\delta^{2}H$  (see Figure 6). although they showed a larger increase in H<sub>2</sub>O concentration than the other tested 821 822 containers. This indicates that the early increase in concentration (ppm) during storage 823 does not significantly affect isotopic composition in the first 24h of storage. From the 824 third day onward, significant shifts in δ<sup>2</sup>H are observed, particularly at 4 °C and 40 °C, 825 while at ambient temperature the isotopic composition remains stable (see Figure 6). Moreover, the stored vapor originates from a 100% water-saturated source, and the 826 827 H<sub>2</sub>O concentration in the vapor exceeds 20000 ppm; therefore, during the first days, 828 potential diffusive exchange with the atmosphere does not appreciably affect the 829 isotopic signal, an effect that becomes evident only from the third day onward, as 830 shown by our results. These findings are consistent with those reported by Magh et al. 831 (2022). In contrast, the two types of bags tested show significant variations in the 832 isotope values.

We provide potential explanations for the observed differences in isotope values between the bag-type containers and the glass bottles:

i. Differences in the sampling of water vapor.

When sampling water vapor in bottles, two syringes are entered through the septum, an inlet and an outlet for the equilibrated water vapor. Once the sampling time is achieved, both syringes are disconnected. In contrast, both types of bags are simply filled with equilibrated air through one syringe. It might be that the throughflow-sampling provides a more stable isotope signature, because the time for throughflow is longer compared to the filling time of the bags (12 minutes per sample vs. 6 minutes per sample).

ii. Isotopic exchange with container material.

As previously shown (Herbstritt, 2023), an isotope exchange between the material of the aluminum bags with the sample air was observed. The worse performance of the bags might be due to this effect, which arguably does not occur for the 250 ml glass bottles. The effect also seems to be less pronounced for the FlexFoil bags. Hence, the production of the aluminum bags might have an influence, if they are not pre-treated.

- In our study, 1 I FlexFoil® bags were the second-best option, with a score of 74% (see
- Figure 7). They performed well in leak-tightness tests but showed larger isotopic
- deviations than glass bottles, especially at ambient temperature. Additionally, their 852
- higher cost and larger volume hinder transport logistics and durability when many 853
- samples are needed. Another important aspect is reuse, which requires labor-intensive 854
- cleaning to reduce memory effects.
- For example, Dahlmann et al. (2025) used affordable multilayer 1 I gas bags with 856
- modified valves to store water vapor and obtained good results with new bags. 857
- However, upon reuse, they recommended rinsing each bag 10 times with dry air and
- restricting each bag to isotopic signatures similar to those from its first use. They also
- identified a storage time-dependent memory effect that can be mitigated by rinsing but 860
- limits bag uses to relatively narrow isotopic ranges. With broad signatures and
- consecutive sampling, reliability is not guaranteed, and further testing is advised. This 862
- supports the view that, in field campaigns involving isotopic labeling (tracer 863
- experiments), reusing bags can induce a strong memory effect, while using new bags 864
- improves performance but may be unsustainable due to cost.
- Finally, 500 ml Aluminum-Zip bags showed the largest isotopic deviations in stored 866
- vapor, only moderate leak-tightness with more pronounced increases after three days,
- and despite their low cost, are effectively single-use with a high risk of puncture during
- transport. Therefore, using this container type compromises data quality. 869
- The differential performance indicates that leak-tightness and the isotopic composition
- of water vapor depend on mechanisms related to container material, closure and
- sealing system, headspace state, and temperature sensitivity. Therefore, to ensure a
- reliable isotopic signature during the first day under variable temperature conditions, 873
- 250 ml glass bottles remain the reference option. Their reusability, low cost,
- reasonable isotopic stability, and suitability for transporting large sample numbers
- make them a practical and effective solution for field sampling campaigns.

#### 877 4.2 Protocol and optimal container for water vapor sampling and storage

Methods for measuring the stable isotopes of water are complex and constantly being 879 improved. Water vapor sampling is a new way to overcome extraction-based methods. 880 However, it requires proper implementation and rigorous, efficient sampling 881 procedures. To reduce uncertainties in water vapor measurements, we propose a refined method and protocol that control the factors most affecting the isotopic 882 883 signature and ensure data quality. According to our results, it is essential to consider 884 container type, flow rate, storage time, and temperature conditions; this priority has

- also been highlighted by other authors (Van Duren, 2004; Magh et al., 2022; Herbstritt 885
- et al., 2023; Bagheri et al., 2021; Benetti et al., 2017). Our data show that experimental 886
- conditions substantially influence the precision of water vapor isotope measurements,
- especially for  $\delta^2$ H, whereas  $\delta^{18}$ O remains relatively stable. Therefore, we emphasize
- the importance of maintaining controlled temperature conditions to minimize

896 897

901

measurement errors, particularly for  $\delta^2 H$ , due to its greater sensitivity to kinetic and diffusive processes (Horita and Wesolowski, 1994; Lamb et al., 2017; Wei et al., 2022; 891 892 Weng et al., 2024; Wen et al., 2008).

894

Our study presents a protocol for sampling and storing water vapor that specifies the container type, storage time, sampling flow rate, and temperature conditions. We identified optimal sampling conditions using 250 ml glass bottles, with flow rates between 75 and 125 ml/min (100 and 125 ml/min performed best), storage times of up to 24 hours, and storage temperatures of 20-25 °C (see Figures 11-13 and Table 2 for details). The results enabled identification of the optimal sampling system configuration. Glass bottles showed the best performance; however, tightness tests with dry air revealed a time-dependent increase in water vapor concentration in ppm. Therefore, the test was repeated using syringes with different diameters, as piercing the septum of 250 ml bottles was suspected to facilitate water vapor intrusion. The results confirmed a progressive increase in concentration, but diameter and size did not appear to be the cause, since both options behaved similarly. The most plausible explanation is diffusive exchange with ambient air initiated immediately after septum perforation during sampling and measurement, which warrants further investigation. The observed pattern indicates that the increase in ppm is consistent with deviations in  $\delta^2$ H that become more pronounced after 72 hours. Isotopic stability depends on the combination of flow rate, temperature, and storage duration; certain configurations extend the effective exposure of the sample and amplify the deviations, for example, 35 ml/min, extreme temperatures of 4 and 40 °C, and storage longer than three days. Consequently, it is essential not to prolong storage and to operate under controlled parameters that minimize deviations in  $\delta^2$ H; see Table 2.

Our results regarding the time window for high-quality measurements are consistent 915 with other studies: Dahlmann et al. (2025) and Herbstritt et al. (2023) report acceptable 916 results for samples stored up to 24 hours ( $\pm 0.7$  and  $\pm 2.3$  % for  $\delta^2 H$  and  $\pm 0.2$  to  $\pm 0.9$  % 917 for  $\delta^{18}$ O); Magh et al. (2022) indicate that measurements can be made up to 3 days 918 later, emphasizing stability over short periods and noting an increasing bias at 7 days, 919 consistent with our observations. Taken together, both our method and previous ones 920 highlight the importance of measuring the obtained samples within the first 24 hours. 921 This might represent a constraint for obtaining reliable water vapor isotope values 922 using this method. Opposed to this, Havranek et al. (2023) implemented an automated 923 vapor storage system (Soil Water Isotope Storage System) based on 650 ml flasks, 924 designed for long-term storage, with reliable results ( $\pm 0.9\%$  for  $\delta^{18}O$  and  $\pm 3.7\%$  for 925 δ<sup>2</sup>H) for 14 days in the laboratory and up to 32 days in the field, recommending not exceeding 40 days. The longer storage times reported there might be a result of the 926 927 larger bottles used in their study. For handling, transport and in terms of sampling time,

however, these large bottles might be a disadvantage.

This temporal limitation may result from diffusive exchange detected in leak-tightness 930 tests with dry air in 250 ml glass bottles, which promotes vapor loss and isotopic

- fractionation, especially at storage temperatures from 4 to 40 °C, where larger isotopic
- deviations were observed. Therefore, vapor should be measured over short periods
- (<24 h), whereas the method of Havranek and collaborators prioritized an autonomous,
- leak-resistant system to ensure long-term storage.

Storing water vapor under ambient conditions preserves the vapor isotopic signal 936 without refrigeration, leorting the feasibility of this method for sampling campaigns in remote regions or areas with limited infrastructure. The study shows that  $\delta^{18}O$  is 937 938 substantially more stable (deviations < ±1% under all evaluated conditions) and can 939 therefore be used reliably for ecohydrological component analysis, whereas δ<sup>2</sup>H is 940 more sensitive to experimental conditions. These findings confirm the effectiveness of the sampling system, which used reusable 250 ml glass bottles with appropriate 941 942 cleaning as described in the Methods section, and a setup constructed from readily 943 available materials (PTFE tubing, adapters, connectors, syringes, compressed dry air 944 from diving cylinders, and mass-flow controllers). However, prolonged storage and extreme temperatures promote isotopic exchange with ambient air and internal vessel 945 946 surfaces (Sturm and Knohl, 2010; Wei et al., 2022). This effect is particularly 947 pronounced for  $\delta^2$ H, while  $\delta^{18}$ O maintains greater precision (Dahlmann et al., 2025; 948 Wen et al., 2008), a pattern demonstrated by our results. Accordingly, we emphasize 949 the need to strictly control sampling flow rate, duration, and storage temperature. 950 Operationally, optimal conditions - moderate to high flow rates, storage less than 24 951 hours, and stable temperature - enable procedural standardization and improve 952 precision, especially for  $\delta^2$ H. However, under extreme temperatures or when samples 953 cannot be processed within 24 hours, deviations can increase and compromise data

### 4.3 Future prospects

One aspect identified in our study was that a potential source of error in the configuration of the sampling method could be leakage in the septum after syringe perforation. Although seemingly insignificant, this effect could cause vapor loss or infiltration of ambient air, promoting isotopic fractionation and contributing to the observed variability of  $\delta^2$ H, as this value is more sensitive to evaporation processes during storage. This mechanism could explain part of the deviations observed in the longer-term storage experiments.

interpretation (Bagheri Dastgerdi et al., 2021; Magh et al., 2022). Finally, selecting

materials to minimize isotopic memory remains a logistical and technical challenge that

must be addressed to ensure data quality (Weng et al., 2024).

This semi-in situ approach offers an innovative alternative for stable isotope analysis of water, serving as a robust complement to direct field measurements (e.g., CRDS/Picarro). It enables the capture of water vapor in equilibrium with soil and xylem, generating isotopic data comparable to in situ measurements and, when used as a substitute, reduces logistical and operational risks. Unlike traditional methods, it avoids destructive procedures on trees and soil (e.g., cryogenic extraction) and allows

sampling at remote sites without compromising the ecohydrological utility of the data.

Increasing the number of replicates and modestly extending sampling time

substantially reduces isotopic variability and facilitates quality control by enabling the

identification and exclusion of unstable replicates while retaining only those with

consistent values and similar means. The use of 250 ml glass bottles provides a parallel line of control and validation that can substitute for in situ instruments when

they are unavailable and enhance them when they are available.

To mitigate this effect in future applications of this water vapor sampling method using 979 250 ml infusion glass bottles (ND 32), it is recommended to use more resistant septa 980 (e.g., PTFE-coated), integrated valve systems, or a secondary seal after perforation 981 (adhesivo termofusible) to maintain tightness. In addition, including experimental 982 controls (perforated vs. non-perforated bottles) will allow more accurate quantification 983 of this source of error. Implementing these improvements would enhance reproducibility and extend the applicability of the method in field campaigns under 984 variable storage and temperature conditions. 985

### 5 Conclusion

988

991

995

999

1005

1010

The isotopic stability of water vapor is determined by the combined effects of the container, closure system, sampling flow rate, and storage time and temperature. Based on our results, the optimal configuration for water vapor sampling is 250 ml infusion glass bottles (ND 32). This container showed the greatest isotopic stability, outperforming 1 I FlexFoil sample bags and 500 ml Aluminum-Zip bags. Optimal performance is achieved at flow rates of 100 to 125 ml/min under ambient conditions (20 to 25 °C). This setup minimizes error and ensures reliable, repeatable  $\delta^{18}$ O and δ<sup>2</sup>H values. For storage, the time-to-analysis should ideally not exceed 24 hours, as storage longer than 24 hours increases isotopic variability, particularly for  $\delta^2 H$ , underscoring the need to minimize the interval between sampling and analysis. If longer storage times are required, the bottles recommended by Havranek et al. (2023) might be an alternative, despite more difficult handling and longer sampling times required. Although δ<sup>18</sup>O remains stable under nearly all conditions with deviations below 0.6%,  $\delta^2$ H variations greater than ±2.5% can occur from the third day onward, emphasizing the importance of limiting storage time to preserve the isotopic signature. While 24 hours is optimal, prolonged storage times of up to 72 hours (3 days) are possible without substantial loss of precision. In this case, a flow rate of 125 ml/min is acceptable to achieve  $\delta^2H$  precision of  $\pm 1.5$  to  $\pm 4\%$  at 20 to 25 °C or 40 °C.

The proposed protocol might be a valuable addition to the existing approaches for water vapor sampling and water isotope analysis from soils and plants without the need for extracting water from the substrates. As such, it provides the opportunity analyze the water isotope values of trees and soils in a high temporal resolution without the need for extensive destructive sampling. Therefore, the method is somewhere in between in situ water vapor measurements and destructive sampling; ideally combining the advantages of both approaches.

### 1012 Data availability

- All data supporting the findings of this study, including raw and processed water vapor
- isotope measurements, calibration files, and experimental metadata, are openly
- available in the Zenodo repository: Iraheta, A., Malsch-Fröhlich, E., Gerchow, M., and
- Beyer, M. (2025). Water vapor isotope sampling dataset for the study "Technical note:
- Water vapor sampling for the analysis of water stable isotopes in trees and soils -
- optimizing sampling protocols" (Version 1.0). Zenodo.
- https://doi.org/10.5281/zenodo.17667032.

### 1020 Author contribution

- All and EMF designed the study and conducted the laboratory experiments. All
- analyzed the data, wrote the first draft of the manuscript, and implemented the
- revisions. MG developed Python scripts for data analysis and visualization and
- contributed to the manuscript review. MB supervised the study and contributed to the
- review and editing of the manuscript.

### 1026 Competing interests

The authors declare that they have no conflict of interest.

### 1028 Acknowledgements

- We are grateful to the German Academic Exchange Service (DAAD) for the doctoral
- scholarship.

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
