# Peer review of "Technical note: Water vapor sampling for the analysis of water stable isotopes"

_EGUsphere, 2025_

## Referee Comment (RC1)

Review EGUSphere-2025-5295

Technical Note: Water Vapor sampling for the analysis of water stable isotopes in trees and soils – optimizing sampling protocols

The authors provide an extensive set of tests and experiments on already published methods. Their findings nevertheless advance our current setups and validate the findings for the proposed methods, and this paper is therefore generally worth publishing. I have a few concerns addressed below and specific comments following that section. I recommend Major Revision before publication.

Dear authors,
Please carefully re-read and re-write the MM section. This is the most crucial for readers that want to setup their own experiments. However, it now is insufficiently structured and difficult to understand for readers new to these methods. Also, I like that you made a figure (1), but it needs some clarification and a legend as well. See details below.

Please re-read and decide whether to write in present or past tense. Additionally, I did not comment on individual language issues, but substantial language editing by a fluent or native speaker would significantly improve readability.

Consider combining figures and merging them since the manuscript has too many.

I commented this at Figure 13 but it is true for the entire manuscript when comparing 2H and 18O data: I feel this representation is misleading, 2H as a fact always has a larger uncertainty in measurements. Since it is roughly 8 times as sensitive you should adapt the sclae by that factor and therewith enhance the readability and comparativeness between the two isotopes.

The discussion is very long and many stretches are simply a repetition of the results (e.g. LL780 ff). It would greatly benefit from being rewritten and shortened. The results should not be repeated, if necessary, reference the respective section.

Specific comments to the text follow here:
**Abstract**:
L 16: specifiy flow rates; e.g. sampling flow rates or measuring?
L 19: delete "with" and then "deviating"
**Keywords**:
Replace "tightness" with "air-tightness" and "container" with "storage container"?

**Introduction**:
L 49: wrong citation style
LL 52 ff: redundant, delete
LL 81 ff: this paragraph is missing the original publication of Havranek's 2020 publication (**https://doi.org/10.1002/rcm.8783**)about the SWISS system, please add for completeness.
L 87: add information about the length of the storage period.
LL 109 ff: Please rewrite and structure this section more carefully, so that the RQ's gain visibility and it becomes clear to the reader which of these you aim to actually address with this work. Please also rewrite the next paragraph, it reads awkward and partially redundant.

Structure the objectives and number them to increase visibility. Also, the last paragraph: "Innovation" does either need a proper header or rewriting to integrate the Innovation.

**Material and Methods**:
Generally, it would be a good idea to replace the section headers with experiment and number? This way every experiment is described with every step needed to carry it out, and nothing is left out or to imagination as it is now.

Figure 1 and Paragraphs 2.1.1/2.1.2: I don't understand why the 4. Point (ND20 and 1L Tedlar bags) is in parentheses and which the tested material is. Please clarify.
Also, the figure could use a proper legend, probably making it easier to differentiate between material and test. Now it looks like there's 2 main experiments, while from the text it seems there should be 4, so maybe a better color-code and symbol legend would enhance understanding at first glance without having to read everything twice?
Also, I understand the decision criteria, but what if the container that was tightest, didn't score as high on the isotope test? How do you proceed from there? Also, what kind of criterion is 3. "time to analysis"? How are the criteria ranked, is the number 1 the most important and decreasing importance? This should be clarified please.
Also, this section is missing the reference to how long the storage is and what was stored?

Section 2.1.3:
The rationale for syringe size is not explained. Please elaborate.
L 198: which method was developed in this study?
L 200: after the first day of sampling? So 7 + 1 day?
L 201: I don't understand, did you measure the same sample after 3 days and again after 7? Or is this a replica? And if so, why is this only mentioned for 3 and 7 days, did you not also measure after 6h and 1 day?
L 203: yes, I get it, but again, readers do not know this yet.
L 206: but this is not a repetition, this is a new experiment with the syringe size. Please be careful with wording.
L 209: so, the dry air test was not conducted using different flow rates?
L 209 ff: which are the isotopic references?
L 214: please stick to either days or hours, this is getting confusing.
L 216: this flow rate depends on analyzer and type it is not a predefined setting for all Picarros.
L 218: so the reference material was stored at these temperatures, or the samples were after sampling?

Section 2.1.4 should be logically before section 2.1.2
L228: are those the same bottles as the ND32?
L235/236: how were those bags treated before reuse?

Section 2.1.5 again, I think assigning each section to a specific experiment would be beneficial to not have to go back and forth between the sections.
LL 242 ff: this is confusing, I would imagine that you used different flow rates for sampling not for measuring? If so, why did you use larger syringes for measuring, I think the flowrate here would be determined by the laser and therefore be low? Pls elaborate and rewrite.
LL246 ff: so all this is only relevant for experiment number 1? Or 3,4?

Section 2.1.7

L 287: I don't see how (3) "time to analysis" would be a critical condition, it is however related to points 1 and 2 but not a point itself, imo. You can decide to sample more often when aiming for an in situ like approach, but you do not need the data in real time, which makes this approach so beautifully suitable for remote regions, where you might find a great experimental site but no infrastructure. If I misunderstand, I would ask you to clarify in the text.

LL 296 ff: I would argue that isotopic stability should have more weight than 50% since it is THE most important criterion in defining the usability of the methods. If all other criteria in sum add to 50% but the sample is not isotopically stable, it is not useful at all and should not be considered.

Section 2.1.8 and subsections, sorry but this is too many subsections, again, I think restructure would enormously benefit the readability here.

L 310: and a reference to the respective section would be nice here.

LL 311 ff: I think that is not entirely correct. The test conditions only provide an assessment for the tightness of the entire storage container, not the syringe size per se. E.g. with the crimped vials the leakage might be due to the lid not fitting properly while you assess it as a syringe size problem. Please either elaborate if I misunderstood or rephrase so that this is clear.

L350 ff: Is this not redundant? I feel you have described this before in the treatment section 2.1.4 or is this a different treatment?

L348: mas flow controller (MFC)

L353: drawn not withdrawn

L369: this should be section 2.1.9 not 2.1.8 please check the remaining sections for correction after this point.

Section 2.2

L435: the liquid values, not fluid?

Section 2.3

L459: Furthermore…?

**Results**:

Figure 5: the resolution of the figure is very poor, please resubmit with a better resolution and larger axis text etc.

Section 3.1.1 What were the initial sampled $H_2O$ concentrations? Did you test whether the bottle truly contained 0 ppm h2O?

Section 3.1.2

L 494 ff: If you do not show the data I think it would be beneficial to the reader if you did not mention the tests at all. Also it would save some time when this is also deleted from the MM section.

Figure 6 also needs a re-do in resolution and text size etc

Figure 8 also needs a re-do in resolution and text size etc

Section 3.2.2.1

The text does not really reflect the results shown in the figure: for example the statement in line 619 ff in the text you combine flow rate, storage time and condition, while the figure pools all samples into the respective condition. So, the present presentation does not support your conclusion. Please either adapt the text or the figure 9. Also, just from looking at the storage time, there is no real effect even at 72 hours storage (i.e. the box of the boxplots as well as the interquartile range does not look so different from the 6 hour storage..)

Figure 9 s.a. and additionally horizontal lines would benefit readability

Section 3.2.1
Either I overread it or the information about how many replicas per flow rate and standard you have is missing for this section. It would be nice if you could add this information to the boxes of the boxplots and the tables you show in this section.
Section 3.2.2.2
It is interesting to see that the low flow rate influences the measurements so much, one would assume the opposite, do you have an idea why?
Can you please reference the Figure 10 sooner? It is much easier to follow the text with the figure already seen.
Figure 10: s.a. and add horizontal lines, also the yaxis should read either δ2H or δD but not δ2D (the same is true for Figure 11 btw)

Section 3.2.3
What is the corresponding analysis you mean here? The rest of the paragraph reads fuzzy and confusing. Please clarify which analysis is the basis for this conclusion and then draw the conclusions. E.g. it is not obvious that flow rates of 125 ml/min truly is the best suited as the results look very similar for all flow rates except the lowest.
Figure 11 seems redundant and considering that the paper already has too many figures, maybe it would be good to delete this one.
Figure 12 can be turned into a table imo

Figure 13: I feel this representation is misleading, 2H as a fact always has a larger uncertainty in measurements. Since it is roughly 8 times as sensitive you could adapt the sclae by that factor and therewith enhance the readability and comparativeness between the two isotopes. You could then also plot both in one plot and in different hues of the same storage time. This is also true for the previous figures and would save you from having to delete too many figures.

**Discussion:**
The discussion is very long and many stretches are simply a repetition of the results (e.g. LL780 ff). It would greatly benefit from being rewritten and shortened. The results should not be repeated, if necessary, reference the respective section.

LL 833 ff:
Conclusion i: then the argument would be that you should sample the bags longer and use an outlet so that the bag is flushed instead of simply filled. Is that something you did?
LL 893 ff: This paragraph can be shortened into its main message, which is lines 904-906. The rest is repetition of results.
LL 914 ff: I have trouble with that conclusion; your results show stability within the first 72 hours as well as did mine in 2022. Why do you then recommend measurements have to happen in the first 24 h?
LL935 ff: Again, this can be boiled down to the main message which is a recommendation based on the results. It does not need to be this long.
Section 4.3 should be shortened and integrated into section 5 Conclusions.
Section 5 is not a conclusion but a nice summary, which could be used in the abstract. Otherwise, it may be redundant and could be removed.